# Model-based cost-effectiveness estimates of testing strategies for diagnosing hepatitis C virus infection in Central and Western Africa

**Léa Duchesne**[1]*, **Gilles Hejblum**[1], **Richard Njouom**[2], **Coumba Touré Kane**[3], **Thomas d'Aquin Toni**[4], **Raoul Moh**[5,6], **Babacar Sylla**[7], **Nicolas Rouveau**[8], **Alain Attia**[9], **Karine Lacombe**[10]

1 Institut Pierre Louis d'Épidémiologie et de Santé Publique, Sorbonne Université, INSERM, Paris, France, 2 Virology Department, Pasteur Centre of Cameroon, Yaoundé, Cameroon, 3 Laboratoire de Bactériologie Virologie, Centre Hospitalier Universitaire Aristide Le Dantec/ Université Cheikh Anta Diop de Dakar, Dakar, Senegal, 4 Centre de Diagnostic et de Recherches sur le SIDA (CeDReS), Centre Hospitalier Universitaire de Treichville, Abidjan, Côte d'Ivoire, 5 Programme PAC-CI, Abidjan, Côte d'Ivoire, 6 Unité de formation et de recherche de Sciences Médicales, Unité Pédagogique de Dermatologie et Infectiologie, Université Félix Houphouët Boigny, Abidjan, 7 Institut de Médecine et d'Epidémiologie Appliquée (IMEA), Paris, France, 8 International Research and Collaboration unit, Agence Nationale de Recherche sur le Sida et les hépatites virales (ANRS), Paris, France, 9 Service d'Hépatologie, Centre Hospitalier Universitaire de Yopougon, Abidjan, Côte d'Ivoire, 10 Institut Pierre Louis d'Epidémiologie et de Santé Publique, Sorbonne Université, INSERM, AP-HP, Hôpital Saint-Antoine, Service des Maladies Infectieuses et Tropicales, Paris, France

* lea.duchesne@iplesp.upmc.fr

## Abstract

### Background

Whereas 72% of hepatitis C virus (HCV)-infected people worldwide live in low- and middle-income countries (LMICs), only 6% of them have been diagnosed. Innovative technologies for HCV diagnosis provide opportunities for developing testing strategies more adapted to resource-constrained settings. However, studies about their economic feasibility in LMICs are lacking.

### Methods

Adopting a health sector perspective in Cameroon, Cote-d'Ivoire, and Senegal, a decision tree model was developed to compare 12 testing strategies with the following characteristics: a one-step or two-step testing sequence, HCV-RNA or HCV core antigen as confirmative biomarker, laboratory or point-of-care (POC) tests, and venous blood samples or dried blood spots (DBS). Outcomes measures were the number of true positives (TPs), cost per screened individual, incremental cost-effectiveness ratios, and nationwide budget. Corresponding time horizon was immediate, and outcomes were accordingly not discounted. Detailed sensitivity analyses were conducted.

### Findings

In the base-case, a two-step POC-based strategy including anti-HCV antibody (HCV-Ab) and HCV-RNA testing had the lowest cost, €8.18 per screened individual. Assuming a lost-

**Funding:** The author(s) received no specific funding for this work.

**Competing interests:** K. Lacombe reports personal fees and non-financial support from GILEAD, personal fees and nonfinancial support from ABBVIE, personal fees and non-financial support from JANSSEN, grants, personal fees and non-financial support from MSD, outside the submitted work. This does not alter our adherence to PLOS ONE policies on sharing data and materials.

to-follow-up rate after screening > 1.9%, a DBS-based laboratory HCV-RNA after HCV-Ab POC testing was the single un-dominated strategy, requiring an additional cost of €3653.56 per additional TP detected. Both strategies would require 8–25% of the annual public health expenditure of the study countries for diagnosing 30% of HCV-infected individuals. Assuming a seroprevalence > 46.9% or a cost of POC HCV-RNA < €7.32, a one-step strategy based on POC HCV-RNA dominated the two-step POC-based strategy but resulted in many more false-positive cases.

## Conclusions

POC HCV-Ab followed by either POC- or DBS-based HCV-RNA testing would be the most cost-effective strategies in the study countries. Without a substantial increase in funding for health or a dramatic decrease in assay prices, HCV testing would constitute an economic barrier to the implementation of HCV elimination programs in LMICs.

## Background

Despite the advent of very effective direct-acting antivirals (DAAs) for treating chronic hepatitis C virus (HCV) infection, the rate of treatment initiation remains low worldwide [1]. One of the main barriers to HCV care is the limited access to testing facilities. Indeed, only an estimated 20% of HCV-infected individuals have been diagnosed globally [1]. In low- and middle-income countries (LMICs), which support the highest burden of HCV-infected individuals, this estimation drops to 6% [1]. Both technical and economic reasons explain this situation.

First, the current reference techniques for diagnosing HCV–anti-HCV antibody (HCV-Ab) screening, followed by confirmation of chronic infection by HCV-RNA testing–require substantial infrastructure and specialized technicians that are scarce in LMICs, especially in rural areas. This situation has favoured the centralization of testing facilities [2]. Moreover, due to the long time-to-result of laboratory assays, both tests cannot be performed on the same visit. Consequently, few individuals are able to access testing and, among those who test HCV-Ab positive, many do not undergo viraemia confirmation.

Second, although this two-step design reduces the procedure's overall cost by avoiding useless HCV-RNA assays being performed, LMICs populations often cannot afford diagnosis costs. Indeed, whereas the prices of DAAs have dramatically decreased in the last few years–in 2017, about 60% of the HCV-infected population worldwide were living in countries with the possibility to access generic DAAs costing about US$100 [3], the cost of HCV tests remains high in most countries [2]. The results of a recent modelling study on 67 LMICs stress the burden represented by HCV testing on global HCV care budgets, especially now that DAAs prices have decreased [4]: reducing the cost of HCV-RNA testing from US$115 to US$20 would decrease from US$24.3 billion to US$16 billion the budget required, within the perspective of universal health coverage, to achieve the screening and treatment coverage goals set by the WHO for reaching HCV elimination. Moreover, even when considering the lowest currently available prices for both treatment and HCV-RNA testing (i.e., US$105 and US$20, respectively), the authors estimated that HCV diagnostics would account for 58% of the overall intervention cost (versus 42% for DAAs).

Given these issues, the World Health Organization (WHO) has prioritized identifying the best strategies for enhancing chronic hepatitis C diagnosis in LMICs [1,5]. Several innovative

technologies could help achieving this objective, by scaling up access to existing infrastructure or creating alternative diagnostic strategies that are better adapted to LMICs. First, while still not routinely included in virology laboratories, detecting HCV core antigen (HCV-cAg) is easier and less expensive than HCV-RNA testing [6,7]. In addition to lowering testing costs, connecting remote and marginalized populations to testing facilities is crucial. Unlike venous blood samples, dried blood spots (DBS) collected on filter paper can be transported from the field without a refrigeration system, stored at room temperature for weeks and then tested with conventional laboratory assays. Recent studies have reported the diagnostic performance of HCV-Ab, HCV-RNA, and HCV-cAg assays on DBS [8–11]. Conversely, point-of-care (POC) tests–portable devices able to perform assays near or at the site of patient care in a short time (5 to 45 minutes)–are designed to bring diagnostics directly to decentralized settings. While many rapid diagnostic tests for HCV-Ab have been available for years, POC HCV-RNA assays have been marketed only recently [12,13].

Implementing field studies that assess the performance and affordability of all the above-described diagnostic tools would be cumbersome and expensive. Therefore, model-based cost-effectiveness analyses (CEAs) constitute an attractive approach to comparing their respective features and feasibility in LMICs. Despite the importance of HCV diagnosis within the HCV care cascade, most recent CEAs dealing with hepatitis C have focused only on DAAs. Those including the issue of HCV testing aimed to determine, considering the high prices of DAAs, how broadly HCV treatment could be scaled up while remaining cost-effective, for instance by comparing risk-based testing with universal testing, or no screening at all [14–19]. Besides, the vast majority of these studies took place in high-income countries [20]. The few CEAs that compared different combination of HCV testing tools have mainly focused on integrating HCV-cAg testing in clinical practice [21–24]. To our knowledge, no study devoted to the comparison of the wide range of currently available HCV diagnostics has been conducted in the context of LMICs. Therefore, the present study was undertaken in order to compare the cost-effectiveness, and the associated budget, of implementing different combination of those tools in the general population of Western and Central sub-Saharan countries.

## Methods

The study findings are reported in line with the Consolidated Health Economic Evaluation Reporting Standards statement [25].

### Diagnostic strategies

Twelve diagnostic strategies based on different combinations of HCV-Ab, HCV-RNA, and HCV-cAg -based technologies were compared (Table 1). Two-step strategies (from $S_{ref}$ [*Lab HCV-Ab (venepuncture)* → *Lab HCV-RNA (venepuncture)*] to $S_9$ [*POC HCV-Ab* → *Lab HCV-cAg (DBS)*]) included screening and, in individuals who previously tested positive for HCV-Ab, viraemia testing whereas single-step strategies ($S_{10}$ [*Lab HCV-cAg (venepuncture)*] to $S_{12}$ [*POC HCV-RNA*]) were only based on viraemia testing. $S_{ref}$ [*Lab HCV-Ab (venepuncture)* → *Lab HCV-RNA (venepuncture)*]) model the reference strategy for diagnosing HCV infection in the three study countries.   Model structure and assumptions

Fig 1 presents the decision trees developed for modelling two-step and one-step strategies (which were adapted from a previous model built to simulate HCV testing strategies in a population of people who use injecting drugs in Dakar, Senegal [26]). Considering that a high part of LMIC populations is rural and may have difficulties accessing laboratories, strategies whose

**Table 1. Description of the HCV testing strategies evaluated in the cost-effectiveness analysis.**

| Strategy | Strategy label | Description |
|---|---|---|
| $S_{ref}$ | *Lab HCV-Ab (venepuncture) → Lab HCV-RNA (venepuncture)* | HCV-Ab testing followed by HCV-RNA testing, both conducted in laboratory on blood samples collected by venepuncture |
| $S_2$ | *Lab HCV-Ab (DBS) → Lab HCV-RNA (DBS)* | HCV-Ab testing followed by HCV-RNA testing, both conducted in laboratory on DBS samples |
| $S_3$ | *POC HCV-Ab → Lab HCV-RNA (venepuncture)* | HCV-Ab testing performed with a POC test followed by HCV-RNA testing conducted in laboratory on blood samples collected by venepuncture |
| $S_4$ | *POC HCV-Ab → Lab HCV-RNA (DBS)* | HCV-Ab testing performed with an POC test followed by HCV-RNA testing conducted in laboratory on DBS samples |
| $S_5$ | *POC HCV-Ab → POC HCV-RNA* | HCV-Ab testing performed with a POC test followed by HCV-RNA testing performed with a POC test |
| $S_6$ | *Lab HCV-Ab (venepuncture) → Lab HCV-cAg (venepuncture)* | HCV-Ab testing followed by HCV-cAg testing, both conducted in laboratory on blood samples collected by venepuncture |
| $S_7$ | *Lab HCV-Ab (DBS) → Lab HCV-cAg (DBS)* | HCV-Ab testing followed by HCV-cAg testing, both conducted in laboratory on DBS samples |
| $S_8$ | *POC HCV-Ab → Lab HCV-cAg (venepuncture)* | HCV-Ab testing performed with a POC test followed by HCV-cAg testing conducted in laboratory on blood samples collected by venepuncture |
| $S_9$ | *POC HCV-Ab → Lab HCV-cAg (DBS)* | HCV-Ab testing performed with a POC test followed by HCV-cAg testing conducted in laboratory on DBS samples |
| $S_{10}$ | *Lab HCV-cAg (venepuncture)* | HCV-cAg testing on blood samples collected by venepuncture |
| $S_{11}$ | *Lab HCV-cAg (DBS)* | HCV-cAg testing on DBS samples |
| $S_{12}$ | *POC HCV-RNA* | HCV-RNA testing performed with a POC test |

Abbreviations: Ab, antibody; cAg, core antigen; DBS, dried blood spot; HCV, hepatitis C virus; HCV-Ab, anti-HCV antibody; HCV-cAg, HCV core antigen; HCV-RNA, HCV ribonucleic acid; lab, laboratory; POC, point of care; RNA, ribonucleic acid; lab, laboratory; RNA, ribonucleic acid; S, strategy; Ven, Venepuncture.

first step is based on POC or DBS testing ("decentralized strategies") may reach more people than strategies using a venepuncture-based laboratory test ("centralized strategies"). This assumption was modelled by introducing the ability to modify the uptake rate of each strategy. For the same reason, patients with a positive HCV-Ab screening test may not be able to return to the laboratory to undergo a viraemia test, as required for venepuncture-based laboratory testing. Varying rates of lost to follow-up (LTFU) after screening (i.e., when patients with a positive HCV-Ab screening test do not undergo viraemia testing) were therefore allowed for two-step strategies involving a venepuncture-based laboratory test that confirms viraemia ($S_{ref}$ [*Lab HCV-Ab (venepuncture) → Lab RNA (venepuncture)*], $S_3$ [*POC HCV-Ab → Lab HCV-RNA (venepuncture)*], $S_6$ [*Lab HCV-Ab (venepuncture) → Lab cAg (venepuncture)*] and $S_8$ [*POC HCV-Ab → Lab cAg (venepuncture)*]). A DBS sample contains various spots of blood. Therefore, both screening and viraemia confirmation can be performed on the same sample, if necessary. Moreover, like POC, DBS collection can be done by a minimally trained individual in a few minutes. Thus, it was considered that LTFU could not occur in two-step strategies including DBS-based testing ($S_2$ [*Lab HCV-Ab (DBS) → Lab HCV-RNA (DBS)*], $S_4$ [*POC HCV-Ab → Lab HCV-RNA (DBS)*], $S_7$ [*Lab HCV-Ab (DBS) → Lab HCV-cAg (DBS)*] and $S_9$ [*POC HCV-Ab → Lab HCV-cAg (DBS)*]).

## Outcomes, time horizon, and perspective

The decision trees were used to estimate the expected cost per screened individual and number of identified true positive (TP) cases, referring to HCV-infected individuals who tested positive

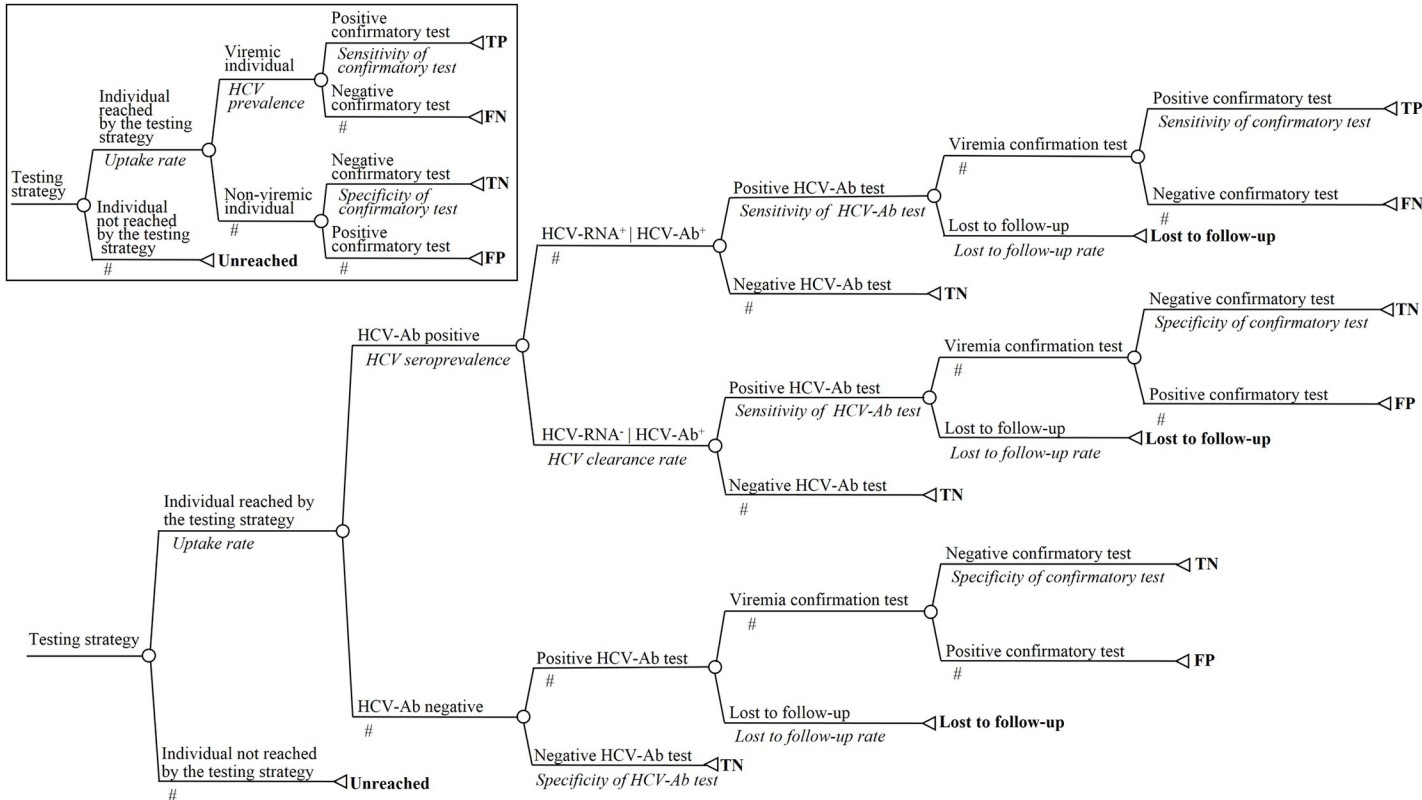

**Fig 1. Structure of the decision trees used to model the studied strategies.** Main panel: tree used for two-step strategies. Inset: tree used for single-step strategies. Chance nodes and terminal nodes are represented by circles and triangles, respectively. The corresponding label and probability of occurrence appear above and below the branch, respectively, for each branch of the tree. The sum of a branch's probabilities of each chance node must equal 1; the # symbol corresponds to 1-probability of the alternative branch. The final effectiveness outcome of a decision pathway is shown to the right of each corresponding terminal node; the associated cost was also estimated. For single-step strategies, the HCV prevalence is equal to the product of the HCV seroprevalence and the HCV clearance rate. Abbreviations: Ab, antibody; HCV, hepatitis C virus; FN, false negative; FP, false positive; LTFU, lost to follow-up; TN, true negative; TP, true positive.

with a viraemia test, for each strategy. The number of true negatives (TNs), false negatives (FNs), false positives (FPs), and diagnostic accuracy of each strategy were also reported.

A one-time screening based on a mass population approach was considered. The time horizon of all effectiveness outcomes investigated in this study was thus immediate (number of individuals screened, TPs, TNs. . .), and therefore no discounting rate was applied to health outcomes and associated costs. Such a time horizon questioned the adoption of a societal perspective in the cost-effectiveness analysis, especially when considering indirect costs: indeed, the relative short-term losses of work productivity between diagnosis strategies were likely uninformative while potential patient time and travel costs for diagnosis inherently constitute intangible costs that cannot be accurately estimated. Therefore, even if patient travel time and associated costs were indirectly considered since sensitivity analyses explored the impact of loss to follow-up, the perspective adopted in the present cost-effectiveness analysis should be considered as a perspective from the health sector.

## Model inputs

Table 2 shows the values of base-case parameters and their respective sources, as well as the ranges and distributions used in the sensitivity analyses.

**Table 2. Values, ranges, and distributions of the model parameters used in the base-case and sensitivity analyses.**

| Variables | Base case | Range considered (sensitivity analyses) | Distribution type (PSA) | Reference |
|---|---|---|---|---|
| **Health outcome and setting, %** | | | | |
| HCV seroprevalence | 0.039 | 0.01–0.60 | Triangular | [27] |
| HCV clearance rate | 0.30 | † | † | [28] |
| Screening uptake rate of centralized strategies | 1 | 0.05–0.60 | † | Assumption |
| Screening uptake rate of decentralized strategies | 1 | 0.30–0.80 | † | Assumption |
| Lost-to-follow-up testing rate between the first- and second-line tests for strategies using a venepuncture-based laboratory test to confirm viraemia | 0 | 0–0.50 | Triangular | Assumption |
| **Test performance, %** | | | | |
| *Sensitivity* | | | | |
| Laboratory HCV-Ab test on serum samples | 0.995 | 0.930–1.000 | Triangular | Product insert§ |
| Laboratory HCV-Ab test on DBS | 0.974 | 0.943–0.988 | Triangular | [29] |
| HCV-Ab POC test | 0.995 | 0.989–0.998 | Triangular | [30] |
| Laboratory HCV-RNA test on serum samples | 0.999 | 0.995–1.000 | Triangular | Product insert¶ |
| Laboratory HCV-RNA test on DBS | 0.980 | 0.950–0.990 | Triangular | [9] |
| HCV-RNA POC test | 0.955 | 0.845–0.994 | Triangular | [31] |
| Laboratory HCV-cAg test on serum samples | 0.934 | 0.901–0.964 | Triangular | [6] |
| Laboratory HCV-cAg test in laboratory on DBS | 0.767 | 0.667–0.850 | Triangular | [11] |
| *Specificity* | | | | |
| Laboratory HCV-Ab test on serum samples | 0.990 | 0.930–1.000 | Triangular | Product insert§ |
| Laboratory HCV-Ab test on DBS | 0.996 | 0.985–0.999 | Triangular | [29] |
| HCV-Ab POC test | 0.998 | 0.996–0.999 | Triangular | [30] |
| Laboratory HCV-RNA test on serum samples | 0.997 | 0.990–1.000 | Triangular | Product insert¶ |
| Laboratory HCV-RNA test on DBS | 0.980 | 0.950–0.990 | Triangular | [9] |
| HCV-RNA POC test | 0.981 | 0.934–0.998 | Triangular | [31] |
| Laboratory HCV-cAg test on serum samples | 0.988 | 0.974–0.995 | Triangular | [6] |
| Laboratory HCV-cAg test on DBS | 0.973 | 0.840–1.000 | Triangular | [11] |
| **Costs, €** | | | | |
| Laboratory HCV-Ab test | 17.5 | 14–23 | ‡‡ | ‡‡ |
| HCV-Ab POC test | 7.6 | +/- 50% | Normal | ‡‡ |
| Laboratory HCV-RNA test | 69.9 | 45–95 | §§ | ‡‡ |
| HCV-RNA POC cartridge | 13.68 | 9.88–13.68 | Triangular | [32] |
| Healthcare worker time for HCV-RNA POC test | 0.6 | 0.4–0.8 | Triangular | ‡‡ |
| Laboratory HCV-cAg test | 34.3 | 22–46 | ¶¶ | ‡‡ |
| DBS sampling | 2.9 | +/- 50% | Normal | ‡‡ |
| DBS transportation from POC to laboratory | 3 | +/- 50% | Normal | ‡‡ |

Abbreviations: DBS, dried blood spot; HCV, hepatitis C virus; HCV-Ab, anti-HCV antibody; HCV-cAg, HCV core antigen; HCV-RNA, HCV ribonucleic acid; POC, point of care; PSA, probabilistic sensitivity analysis.

†The base-case value was used in the corresponding sensitivity analysis.

§Vitro anti-HCV Assay (Ortho Clinical Diagnostics).

¶Abbott Real Time HCV Viral Load (Abbott Diagnostics).

††Randomly chosen among three possible values: 14.48, 15.24, 23 (cf. Table 3).

‡‡Personal communication from the study sites.

§§Randomly chosen among three possible values: 95.3, 45.7, 68.7 (cf. Table 3).

¶¶Randomly chosen among three possible values: 45.7, 22.8, 34.3 (cf. Table 3).

**Effectiveness of diagnostic tests.** The performances (sensitivity and specificity) of all assays, on both type of samples for venepuncture-based tests, were retrieved from the literature. The ranges used for the deterministic sensitivity analyses of the corresponding parameters are based on the confidence intervals of these estimates.

**Costs.** Costs were assessed from a health sector perspective and, therefore, only direct medical costs associated to HCV diagnostic were included. The costs of laboratory testing and POC tests were obtained for the year 2018 using data collection in three reference laboratories in Cameroon (Centre Pasteur, Yaoundé), Côte d'Ivoire (Yopougon Hospital, Abidjan), and Senegal (Fann Hospital, Dakar). Costs comprised those of reagents and operating costs, including corresponding human resources. The cost of the HCV-RNA POC test was retrieved from the literature. For the base-case analysis, the median of the available prices was used. All values were expressed in 2018 euros (€1 = FCFA655.96, €1 = US\$0.86) and not discounted since an immediate time horizon was considered.

## Analysis

**Base-case and sensitivity analyses.** Strategies which identified fewer TPs than less expensive strategies correspond to dominated strategies. Beginning with the lowest cost strategy, un-dominated alternatives were compared with the next most costly un-dominated strategy to calculate the incremental cost-effectiveness ratio (ICER), the ratio of the difference in costs between two strategies to the corresponding difference in the number of TPs identified.

The impact of the parameters' uncertainties on the estimated costs, health gains, and cost-effectiveness ranking was explored using univariate sensitivity analyses conducted on all variables. Except those based-on HCV-cAg, the diagnostic performance estimates used in the base-case analysis for the different assays were close (Table 2). As the reported differences in the strategies' effectiveness may therefore only reflect potential sampling fluctuations in the assessment of those estimates, a scenario assuming identical sensitivity (98%) and specificity (99%) values for all assays except HCV-cAg-based assays was explored. Reducing the HCV diagnostic procedure to one single visit was identified as a potential lever for scaling up HCV testing [33,34]. A POC-based one-step strategy ($S_{12}$ [*POC HCV-RNA*]) might increase the feasibility of diagnostics since the procedure duration would decrease. Therefore, the cost threshold of an HCV-RNA POC cartridge at which $S_{12}$ [*POC HCV-RNA*] would become the most cost-effective strategy was also estimated. A probabilistic sensitivity analysis based on a micro-simulation of 10,000 iterations was conducted. A corresponding cost-effectiveness acceptability curve (CEAC) was created, showing the probability of a strategy being cost-effective as a function of the willingness-to-pay (WTP) for detecting a TP. To our knowledge, no cost-effectiveness threshold has been defined for health outcomes similar to the one used in the present study (i.e., number TP identified). One of the most commonly used cost-effectiveness thresholds for LMICs considers as cost-effective interventions that cost less than 1 to 3 times the national gross domestic product (GDP)-per-capita per quality-adjusted life years (QALY) gained in the studied country [35,36]. Although enhancing diagnosis contributes to preventing HCV transmission–by enabling identified HCV-infected individuals to take preventive measures–, most health gains in terms of QALY would result from treatment. It can therefore be hypothesized that the cost-effectiveness of diagnostic procedures should be assessed according to a much lower threshold than that used for curative interventions.

**Budget analyses.** The WHO estimated that, to end the HCV epidemic, 30% of HCV-infected individuals should be diagnosed by 2020 and 90% by 2030 [1]. To evaluate the affordability of the strategies that were consistently dominant in the CEA, the costs of achieving the WHO targets by implementing those strategies in the general population (aged 15 and over) of

**Table 3. Values of the model parameters used for the budget analyses.**

| | Cameroon | Côte d'Ivoire | Senegal | Reference |
|---|---|---|---|---|
| **HCV seroprevalence (%)** | 4.9 | 2.2 | 1.0 | [37] |
| **Demographic indicators** | | | | |
| Population ($10^6$ individuals aged 15 and over), year 2018 | 14.19 | 14.38 | 9.33 | [38] |
| Urban population (%) | 47.9 | 51.2 | 43.3 | [39] |
| **Economic statistics** | | | | |
| Estimated GDP per capita (€), year 2018 | 1,266 | 1,516 | 975 | [40] |
| Estimated national GDP ($10^9$ €), year 2018 | 31.5 | 38.8 | 15.9 | [40] |
| Health expenditures (% of GDP), year 2015 | 0.7 | 1.2 | 1.4 | [41] |
| **Testing costs (€)** | | | | |
| HCV-Ab testing in laboratory | 14.48 | 15.24 | 23.00 | † |
| HCV-Ab POC test | NA‡ | NA‡ | 7.60 | † |
| HCV-RNA testing in laboratory | 95.30 | 45.70 | 68.60 | † |
| HCV-RNA POC test | 13.68 | 13.68 | 13.68 | † |
| HCV-cAg testing in laboratory | 45.70 | 22.90§ | 34.30§ | † |

Abbreviations: GDP, gross domestic product; HCV, hepatitis C virus; HCV-Ab, anti-HCV antibody; HCV-cAg, HCV core antigen; HCV-RNA, HCV ribonucleic acid; NA, not available; POC, point of care.

†Personal communication from the study sites.

‡Since no data were available regarding the costs for the HCV-Ab POC test in Cameroon and Côte d'Ivoire, the corresponding cost in Senegal was used for the budget analyses.

§To our knowledge, HCV-Ag testing had not already been implemented in the three case countries at the time of the present study. Therefore, based on experts' forecasts, a hypothetical cost of HCV-Ag testing was assumed as half the cost of HCV-RNA testing.

each study countries were estimated on the basis of the mean cost per screened individuals found by the CEA model for each country. The parameters used for this analysis are presented in Table 3.

All simulations were performed with TreeAge Pro software (© 2018, TreeAge Software, Inc. Williamstown, MA, USA). Figures were drawn with R 3.6.1 (the corresponding R codes can be found at the following address: https://github.com/LeaDuchesne/Cost_effectiveness_ HCV_testing_Central_Western_Africa), except for Fig 1 which was drawn with Microsoft PowerPoint 2013.

## Results

### Base-case analysis

The cost, effectiveness, and cost-effectiveness of each strategy are reported in Table 4. Neither any uptake loss nor any LTFU was considered. $S_5$ [*POC HCV-Ab → POC HCV-RNA*] was the least expensive strategy with a cost per individual screened of €8.18. Compared to $S_5$ [*POC HCV-Ab → POC HCV-RNA*], all strategies were dominated, except $S_3$ [*POC HCV-Ab → Lab HCV-RNA (venepuncture)*] with an ICER of €1,895.29 per additional TP identified. The specificity of both strategies was similar, but $S_3$ [*POC HCV-Ab → Lab HCV-RNA (venepuncture)*] displayed a better sensitivity than $S_5$ [*POC HCV-Ab → POC HCV-RNA*]: 99.4% (271/273) versus 95.02% (259/273), respectively.

### Sensitivity analyses

S1 Fig shows the percentage of HCV-infected individuals identified in the target population, assuming different LTFU rates in the two-step strategies using laboratory tests on venous

**Table 4. Base-case estimates of cost, effectiveness, and cost-effectiveness of testing strategies for detecting chronic hepatitis C cases.**

| Strategy | Cost / screened individual (€) | Number of true positive cases / 10,000 screened individuals | ICER (€ / additional true positive case detected) | Cost / true positive case detected (€) | Diagnostic accuracy (%)† | False positives / 10,000 screened individuals | False negatives / 10,000 screened individuals | True negatives / 10,000 screened individuals |
|---|---|---|---|---|---|---|---|---|
| S$_5$: *POC HCV-Ab → POC HCV-RNA* | 8.18 | 259 | | 316 | 99.84 | 3 | 14 | 9,724 |
| S$_8$: *POC HCV-Ab → Lab HCV-cAg (venepuncture)* | 9.00 | 254 | ¶ | 355 | 99.79 | 2 | 19 | 9,725 |
| S$_9$: *POC HCV-Ab → Lab HCV-cAg (DBS)* | 9.24 | 208 | ¶ | 443 | 99.32 | 4 | 65 | 9,723 |
| S$_3$: *POC HCV-Ab → Lab HCV-RNA (venepuncture)* | 10.45 | 271 | 1,895.29 | 385 | 99.98 | 0 | 2 | 9,727 |
| S$_4$: *POC HCV-Ab → Lab HCV-RNA (DBS)* | 10.69 | 266 | ¶ | 401 | 99.90 | 3 | 7 | 9,724 |
| S$_{12}$: *POC HCV-RNA* | 14.28 | 261 | ¶ | 571 | 98.03 | 185 | 12 | 9,542 |
| S$_6$: *Lab HCV-Ab (venepuncture) → Lab HCV-cAg (venepuncture)* | 19.16 | 254 | ¶ | 755 | 99.78 | 3 | 19 | 9,724 |
| S$_{ref}$: *Lab HCV-Ab (venepuncture) → Lab HCV-RNA (venepuncture)* | 20.88 | 271 | ¶ | 770 | 99.98 | 1 | 2 | 9,726 |
| S$_7$: *Lab HCV-Ab (DBS) → Lab HCV-cAg (DBS)* | 24.83 | 204 | ¶ | 1224 | 99.27 | 4 | 69 | 9,723 |
| S$_2$: *Lab HCV-Ab (DBS) → Lab HCV-RNA (DBS)* | 26.32 | 261 | ¶ | 1015 | 99.84 | 3 | 12 | 9,724 |
| S$_{10}$: *Lab HCV-cAg (venepuncture)* | 34.30 | 255 | ¶ | 1345 | 98.65 | 117 | 18 | 9,610 |
| S$_{11}$: *Lab HCV-cAg (DBS)* | 40.20 | 209 | ¶ | 1920 | 96.74 | 263 | 64 | 9,464 |

Abbreviations: Ab, antibody; cAg, core antigen; DBS, dried blood spot; ICER, incremental cost-effectiveness ratio; lab, laboratory; POC, point of care; RNA, ribonucleic acid; S, strategy; Ven, Venepuncture.

†Diagnostic accuracy: cumulated percentage of true positive and true negative cases.

¶Dominated strategy.

blood samples (S$_3$ [*POC HCV-Ab → Lab HCV-RNA (venepuncture)*], S$_6$ [*Lab HCV-Ab (venepuncture) → Lab HCV-cAg (venepuncture)*], S$_8$ [*POC HCV-Ab → Lab HCV-cAg (venepuncture)*], and S$_{ref}$ [*Lab HCV-Ab (venepuncture) → Lab HCV-RNA (venepuncture)*]) to confirm viraemia. As the LFTU rate increases, fewer viraemia tests are performed, leading to a decreased percentage of identified HCV-infected individuals and, to a lesser extent, a decreased cost per screened individual using S$_3$ [*POC HCV-Ab → Lab HCV-RNA (venepuncture)*], S$_6$ [*Lab HCV-Ab (venepuncture) → Lab HCV-cAg (venepuncture)*], S$_8$ [*POC HCV-Ab → Lab HCV-cAg (venepuncture)*], and S$_{ref}$ [*Lab HCV-Ab (venepuncture) → Lab HCV-RNA (venepuncture)*]. Consequently, venepuncture-based laboratory tests became less cost-effective than their DBS-based equivalents (i.e., S$_4$ [*POC HCV-Ab → Lab HCV-RNA (DBS)*], S$_7$ [*Lab HCV-Ab (DBS) → Lab HCV-cAg (DBS)*], S$_9$ [*POC HCV-Ab → Lab HCV-cAg (DBS)*], and S$_2$ [*Lab HCV-Ab (DBS) → Lab HCV-RNA (DBS)*]) above specific LTFU rates. Most importantly,

a change in the dominant strategies was observed for any LTFU rate above 1.9%, with S$_4$ [*POC HCV-Ab → Lab HCV-RNA (DBS)*] becoming more cost-effective than S$_3$ [*POC HCV-Ab → Lab HCV-RNA (venepuncture)*]. Since S$_5$ [*POC HCV-Ab → POC HCV-RNA*] and S$_4$ [*POC HCV-Ab → Lab HCV-RNA (DBS)*] are unaffected by LTFU, the ICER of S$_4$ [*POC HCV-Ab → Lab HCV-RNA (DBS)*] to S$_5$ [*POC HCV-Ab → POC HCV-RNA*] remained constant for any LTFU rate greater than this threshold, with an ICER of €3,689.55 per additional TP detected.

An increase in HCV seroprevalence directly increases the number of viraemia tests performed in two-step strategies, increasing their cost per screened individuals (S2 Fig). With a seroprevalence value greater than 46.9%, S$_{12}$ [*POC HCV-RNA*] became more cost-effective than S$_5$ [*POC HCV-Ab → POC HCV-RNA*]. However, at this threshold, S$_{12}$ [*POC HCV-RNA*] was associated with a substantially higher number of FPs than the other un-dominated strategies (S1 Table). Compared to S$_{12}$ [*POC HCV-RNA*], S$_4$ [*POC HCV-Ab → Lab HCV-RNA (DBS)*] would produce fewer false results but would have an ICER of €4,360 per additional TP detected.

When assuming identical diagnostic performance for all assays except HCV cAg-based assays, S$_5$ [*POC HCV-Ab → POC HCV-RNA*] and S$_{12}$ [*POC HCV-RNA*] were the dominant strategies. However, S$_{12}$ [*POC HCV-RNA*] detected only 3 additional TPs per 10,000 individuals screened compared to S$_5$ [*POC HCV-Ab → POC HCV-RNA*], with a corresponding ICER of about €22,000. Moreover, S$_{12}$ [*POC HCV-RNA*] produced many more FPs than S$_5$ [*POC HCV-Ab → POC HCV-RNA*] (97 and 2, respectively). With the same effectiveness as S$_5$ [*POC HCV-Ab → POC HCV-RNA*] but at a higher cost, S$_4$ [*POC HCV-Ab → Lab HCV-RNA (DBS)*] was dominated by S$_5$ [*POC HCV-Ab → POC HCV-RNA*].

No other parameter had an impact on the strategies' cost-effectiveness ranking in the univariate sensitivity analyses. However, ICERs were sensitive to variations in the cost of viraemia tests (S3 Fig). Also, the results of the sensitivity analysis exploring the impact of variations in the ratio of the screening uptake rate of decentralized strategies to the screening uptake rate of centralized strategies are presented in S1 Text.

A sensitivity analysis exploring various costs of the HCV-RNA POC cartridge indicated that when it cost less than €7.32, S$_{12}$ [*POC HCV-RNA*] would dominate S$_5$ [*POC HCV-Ab → POC HCV-RNA*]. Furthermore, even a slight increase above this threshold value dramatically increases the corresponding ICER of S$_{12}$ [*POC HCV-RNA*] to S$_5$ [*POC HCV-Ab → POC HCV-RNA*].

In the probabilistic sensitivity analysis (Fig 2), 8 of the 12 strategies were not favoured by any of the simulations, regardless of the WTP threshold adopted. Moreover, two of the four remaining strategies—S$_8$ [*POC HCV-Ab → Lab HCV-cAg (venepuncture)*] and S$_{12}$ [*POC HCV-RNA*]—had less than 5% of favourable results for most of the WTP thresholds. In contrast, the rate of favourable simulations for S$_4$ [*POC HCV-Ab → Lab HCV-RNA (DBS)*] and S$_5$ [*POC HCV-Ab → POC HCV-RNA*] highly depended on the WTP value: for a WTP below €1,000, over 90% of the simulations were favourable to S$_5$ [*POC HCV-Ab → POC HCV-RNA*], while above a WTP threshold of €2,400 –about 2 times the average GDP-per-capita in Sub-Saharan countries (€1,340) [42]– the proportion of simulations favouring S$_4$ [*POC HCV-Ab → Lab HCV-RNA (DBS)*] exceeded that favouring S$_5$ [*POC HCV-Ab → POC HCV-RNA*].

## Budget analyses

Table 5 lists the calculated estimates. The costs incurred for achieving the WHO targets of 30% and 90% of HCV-infected individuals diagnosed in each country would constitute a substantial share of the current public health expenditures 8–25% and 23–75%, respectively. Enforcing S$_4$ [*POC HCV-Ab → Lab HCV-RNA (DBS)*] instead of S$_5$ [*POC HCV-Ab → POC HCV-RNA*] would increase the diagnosis budget by 49.1% in Cameroon, 8.3% in Côte d'Ivoire, and 2.2% in Senegal. Regarding their effectiveness, S$_4$ [*POC HCV-Ab → Lab HCV-RNA (DBS)*]

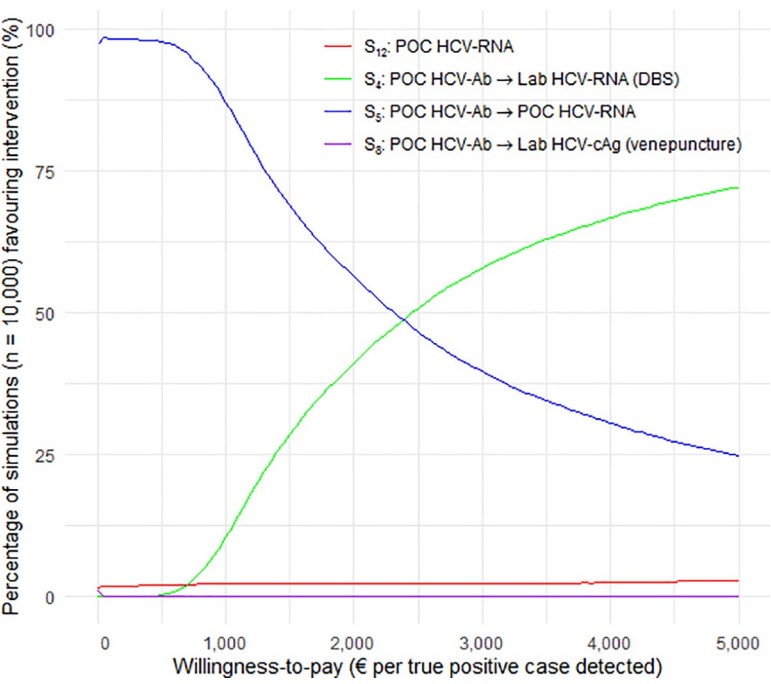

**Fig 2. Cost-effectiveness acceptability curve showing the probability favouring a given strategy according to the willingness-to-pay for detecting a true positive case.** Abbreviations: Ab, antibody; cAg, core antigen; DBS, dried blood spot; lab, laboratory; POC, point of care; RNA, ribonucleic acid; S, strategy; Ven, Venepuncture.

implementation would result in 2.5% more FPs than $S_5$ [*POC HCV-Ab → POC HCV-RNA*] but in half as many FNs (S2 Table).

## Discussion

A decision-analytic model based on data from Cameroon, Côte d'Ivoire, and Senegal was developed to compare the cost-effectiveness of 12 different strategies for diagnosing HCV. In the base-case analysis, strategies combining the screening of HCV-Ab with an POC and the detection of HCV-RNA, either by POC testing ($S_5$ [*POC HCV-Ab → POC HCV-RNA*]) or

**Table 5. Estimated costs for achieving the WHO HCV testing targets by implementing $S_4$ and $S_5$ in Cameroon, Côte d'Ivoire, and Senegal.**

| Country and strategy | Total cost for achieving the diagnosis of 30% of HCV-infected individuals | | | Total cost for achieving the diagnosis of 90% of HCV-infected individuals | | |
|---|---|---|---|---|---|---|
| | $10^6$ € | % GDP | % Public HE | $10^6$ € | % GDP | % Public HE |
| **Cameroon** | | | | | | |
| $S_5$: *POC HCV-Ab → POC HCV-RNA* | 37.3 | 0.12 | 17.0 | 111.9 | 0.36 | 50.7 |
| $S_4$: *POC HCV-Ab → Lab HCV-RNA (DBS)* | 55.6 | 0.18 | 25.2 | 166.7 | 0.53 | 75.6 |
| **Côte d'Ivoire** | | | | | | |
| $S_5$: *POC HCV-Ab → POC HCV-RNA* | 36.1 | 0.09 | 7.8 | 108.3 | 0.28 | 23.3 |
| $S_4$: *POC HCV-Ab → Lab HCV-RNA (DBS)* | 39.1 | 0.10 | 8.4 | 117.2 | 0.30 | 25.2 |
| **Senegal** | | | | | | |
| $S_5$: *POC HCV-Ab → POC HCV-RNA* | 22.9 | 0.14 | 10.3 | 68.7 | 0.43 | 30.9 |
| $S_4$: *POC HCV-Ab → Lab HCV-RNA (DBS)* | 24.4 | 0.15 | 10.9 | 73.1 | 0.46 | 32.8 |

Abbreviations: Ab, antibody; DBS, dried blood spot; GDP, gross domestic product; HCV, hepatitis C virus; HE, health expenditure; lab, laboratory; POC, point of care; RNA, ribonucleic acid; WHO, World Health Organization; S, strategy.

venepuncture-based laboratory testing (S$_3$ [*POC HCV-Ab → Lab HCV-RNA (venepuncture)*]), appeared to be the most cost-effective procedures. However, when two-step strategies comprising a venepuncture-based viraemia test included LTFU, using an HCV-Ab POC test followed by a laboratory HCV-RNA test on DBS (S$_4$ [*POC HCV-Ab → Lab HCV-RNA (DBS)*]) became more cost-effective than S$_3$ [*POC HCV-Ab → Lab HCV-RNA (venepuncture)*], even for a very low LTFU rate (<2%). This result was confirmed by the probabilistic sensitivity analysis, which found that S$_5$ [*POC HCV-Ab → POC HCV-RNA*] and S$_4$ [*POC HCV-Ab → Lab HCV-RNA (DBS)*] were the strategies with the highest number of favourable simulations across a broad range of WTP values. A RNA-based one-step strategy was found un-dominated in some particular situations (i.e., seroprevalence values ≥ 46.9% and a cost of POC HCV-RNA cartridge < €7.32), but was always associated with a relatively high rate of FPs. Because of the lower sensitivity of HCV-cAg detection compared to all HCV-RNA assays, none of the strategies involving this method was found as a cost-effective alternative, and this, in spite of its lower cost compared to HCV-RNA laboratory testing.

The budget analyses indicate that implementing S$_4$ [*POC HCV-Ab → Lab HCV-RNA (DBS)*] instead of S$_5$ [*POC HCV-Ab → POC HCV-RNA*] would represent a modest increase in the diagnosis budget required for reaching the WHO target in Senegal and Côte d'Ivoire (+2.2% and +8.3%, respectively) but a substantial budget increase in Cameroon (+49.1%). This is due to the higher HCV seroprevalence and difference in cost between POC and laboratory HCV-RNA tests in Cameroon when compared to Côte d'Ivoire and Senegal. Nevertheless, choosing S$_4$ [*POC HCV-Ab → Lab HCV-RNA (DBS)*] would halve the number of FNs and produce only 2.5% more FPs; the additional costs incurred by the treatment of FPs would therefore be similar for both strategies. Still, for both strategies, the estimated expenditure required for achieving the WHO 30% goal would represent 8–25% of the State public health budget, an expenditure that these countries cannot cover in the short term.

Indeed, in LMICs, especially in sub-Saharan Africa, domestic governmental health expenditures account for a minor share of the overall current health expenditures [14.5–31.7% in the study countries [43]). This budget is almost entirely dedicated to the national public health care system's operating costs. The leeway for reallocating these governmental funds and promoting new interventions without hindering the proper functioning of fundamental health infrastructures is therefore limited. The remaining health revenue sources are the private sector (51.8%–77.6% [44]) and foreign funding (7.9%–26.3% [45]). Due to the lack of health insurance, household out-of-pocket expenditures constitute at least 70% of private health expenditures. Yet, in the three study countries, completing the HCV diagnosis process—screening and viraemia confirmation—with S$_5$ [*POC HCV-Ab → POC HCV-RNA*] would cost up to 40% of the minimum monthly wage (i.e., €55, €91, and €73 in Cameroon, Côte d'Ivoire, and Senegal, respectively [39]). Moreover, about 30% of their population live below the international poverty line (i.e., with less than €1.2 per day) [39]. Thus, in these settings, even the least expensive strategy identified in this study would represent an unsustainable financial burden for most households.

In those countries, closing the funding gap to reach the WHO targets will therefore probably require negotiations to lower assays' prices and increased mobilization of external funding. In the case of DAAs, the mobilization of economic and commercial facilities (voluntary licenses, compulsory licenses, tiered pricing, etc.) has led to an improvement in the overall affordability of DAAs worldwide [46,47]. Transposing these facilities to hepatitis C diagnostics is just beginning to be explored. For instance, manufacturers and community members have discussed alternative pricing systems, such as offering tiered pricing [32]. For instance, Cepheid offers discounts for an annual minimum purchase of 500,000 tests [32]. However, such a threshold is well above the estimated number of viraemia tests required to reach the WHO 90% target in each of the study countries which did not exceed 300,000 tests. New

products may change in the near future this commercial landscape. For instance, the product development partnerships FIND has recently announced to be seeking partners for commercializing their recently developed HCV-cAg POC test, whose cost of goods sold is currently lower than US$5, in LMICs. Economic evaluations have played a key role in raising awareness of the need for action at the economic level to enable the widespread of DAAs. Such studies on HCV screening and diagnosis remain scarce. Yet, the results of the present study indicate that more than a technical barrier, HCV screening and diagnosis may represent a financial barrier to the expansion of HCV care in LMICs. This suggests that more emphasis should be given to this very first element of the cascade of HCV care in the future economic studies addressing the issue of its scaling- up in LMICs.

This study has several limitations. First, the estimates of the tests' diagnostic performance were derived from studies conducted primarily in high-income countries. Yet, since test performance can decrease in LMICs, the effectiveness of the strategies may have been overestimated. Second, the small differences in effectiveness observed in the base-case can be explained by the slight differences between the performance estimates used in the model; whether these differences reflect real differences in diagnostic performance or only sampling fluctuations is questionable. Therefore, although the CEAC corroborated the dominance of $S_4$ [*POC HCV-Ab → Lab HCV-RNA (DBS)*] and $S_5$ [*POC HCV-Ab → POC HCV-RNA*], future simultaneous field evaluations of the performances of the different tests included in these strategies are required. Third, health gains were not expressed in QALYs but in number of TP cases diagnosed, which may limit the comparability of the present study, as it is not a commonly used outcome in CEAs. This choice was driven by the very objective of the study which aimed to compare many potential combinations of testing tools in order to identify the most suitable ones for implementing large-scale HCV testing in the study countries. Indeed, considering the call for scaling-up HCV screening worldwide, the availability of new HCV diagnostic tools, and the knowledge gap regarding their economic feasibility in LMICs, we deliberately focused on testing and did not consider subsequent care, especially HCV treatment. Fourth, the unit costs used in the model were only collected in three sub-Saharan countries, limiting the generalizability of the results. Moreover, since these costs were derived from only one testing centre in each country, the calculations did not consider the cost variations that may exist within a country. In addition, some institutions did not use all the assays in their daily practice, resulting in missing data that had to be inferred. Hence, costs may have been under- or over-estimated. The analyses conducted to evaluate the impact of this uncertainty showed that ICER estimates were sensitive to variations in the tests' prices. As the example of a decreased cost of HCV-RNA POC cartridges resulting in the dominance of the POC-based one-step strategy ($S_{12}$ [*POC HCV-RNA*]) illustrates, negotiations in the tests' prices would therefore change the cost-effectiveness ranking of the strategies and represents a potential lever for LMIC health officials willing to scale up access to HCV testing. Fifth, this analysis did not consider the potential initial investment that the different strategies would require. Scaling-up laboratory testing requires investments in training and staffing, as well as acquiring heavy equipment costing between €45,000 and €140,000 [48]. Due to their simplicity of use, POC tests will probably require a lower investment in human resources. The GeneXpert device has a lower purchase price (€15,000) [32], but, because of its lower maximum batch size and potential decentralized use, it would probably need to be purchased in higher quantities to cover an entire country's needs, which may incur a higher initial investment than for laboratory machines. Moreover, because some manufacturers propose providing free-on-loan machines in exchange for a given number of purchased assays per year, the initial required investment may vary across countries [32]. Generally, because of the difficulty of assessing the number of pre-existing resources in each country, the corresponding parameters were not

included in the model. Finally, the combination of the following elements suggests that a more selective screening approach (risk-based, cohort-based) might be more appropriate than the mass population screening approach presented in the budget analysis (Table 5): the scarcity of resources, the low HCV seroprevalence in the general population of the study countries (1–4.9%), and the previous identification of risk factors for HCV infection in some countries [49–51]. Nevertheless, Table 5 also indicates that even a selective strategy targeting only 30% of the population would require a substantial increase of public health resources.

In conclusion, this study suggests that two-step decentralized strategies based on POC or DBS testing would be the most cost-effective procedures for HCV testing in the general population of resource-constrained countries with low to moderate seroprevalence. However, documenting some setting-specific key parameters appeared to be critical for choosing the most suitable strategy: future studies are needed to estimate more accurately the relative diagnostic performances of the strategies in routine practice, the cost of HCV RNA tests, the HCV seroprevalence, and LTFU rates in two-step strategies. This study also indicates that, given the current prices of HCV assays, implementing large-scale HCV-testing campaigns would be too expensive to be covered by their health systems. Considering the structure of health funding in these countries, negotiating assay prices and finding supplementary funding for health are therefore required for making such programs affordable in LMICs.

## Supporting information

**S1 Table. Expected cost per screened individual and numbers of true positive, false positive, false negative, and true negative cases per 10,000 screened individuals under the base-case assumptions, excluding a seroprevalence value assumed at 46.9%.** Abbreviations: Ab, antibody; cAg, core antigen; DBS, dried blood spot; lab, laboratory; POC, point of care; RNA, ribonucleic acid; S, strategy; Ven, Venepuncture.
(DOCX)

**S2 Table. Expected cost per screened individual and expected numbers of true positive, false positive, false negative, and true negative cases for achieving the WHO HCV testing targets by implementing the testing strategies $S_5$ and $S_4$ in 31.6% and 30.8%, respectively, of the general population of Cameroon, Côte d'Ivoire, and Senegal.** Abbreviations: Ab, antibody; DBS, dried blood spot; GDP, gross domestic product; HCV, hepatitis C virus; HE, health expenditure; lab, laboratory; POC, point of care; RNA, ribonucleic acid; S, strategy; WHO, World Health Organization.
(DOCX)

**S1 Text. Sensitivity analysis: impact of variations in the ratio of the screening uptake rate of decentralized strategies to the screening uptake rate of centralized strategies.**
(DOCX)

**S1 Fig. Percentage of HCV-infected individuals in the target population diagnosed by the studied two-step strategies according to different lost-to-follow-up rates and the type of samples used.** Each graph shows the results of the sensitivity analysis for strategies based on the same testing sequence, i.e., strategies including the same tests (same biomarker and same test setting: POC versus laboratory) and the same number of steps. The strategies of each pair differ only on the kind of samples used: venous blood samples (red line) or DBS (blue line). All of the model's other parameters were set at their base-case values. Abbreviations: DBS, dried blood spot; lab, laboratory; POC, point of care; RNA, ribonucleic acid; S, strategy.
(TIFF)

**S2 Fig. Cost per screened individual of each studied testing strategy according to different levels of HCV seroprevalence.** All of the model's parameters, except HCV seroprevalence, were set at their base-case values. The dashed vertical line represents the base-case value for HCV seroprevalence (3.9%). The colored lines correspond to the strategies that were consistently dominant in the cost-effectiveness analysis. Abbreviations: DBS, dried blood spot; lab, laboratory; POC, point of care; RNA, ribonucleic acid; S, strategy.
(TIF)

**S3 Fig. Tornado plot indicating the changes in the incremental cost-effectiveness ratio (ICER) of $S_4$ versus $S_5$ resulting from variations of the most influential parameters of the cost-effectiveness analysis.** In each of the explored scenarios (top to bottom), a model parameter's value was changed in comparison to the base-case. The vertical line reflects the ICER of $S_4$ [*POC HCV-Ab → Lab HCV-RNA (DBS)*] versus $S_5$ [*POC HCV-Ab → POC HCV-RNA*] under the base-case assumptions. The blue portion of a bar represents the parameter range from the low uncertainty value to the base-case, while the red portion represents the parameter range from the base-case to the high uncertainty value. For each of these scenarios, the figure indicates how the ICER changed in comparison to the base-case. For sensitivity values of an HCV-RNA POC greater than 0.98, $S_5$ [*POC HCV-Ab → POC HCV-RNA*] became more effective than $S_4$ [*POC HCV-Ab → Lab HCV-RNA (DBS)*], resulting in a negative ICER, designated here by the "$\infty$" symbol. Abbreviations: Ab, antibody; DBS, dried blood spot; EV, expected value; HCV, hepatitis C virus; POC, point of care; RNA, ribonucleic acid; S, strategy.
(TIF)

## Author Contributions

**Conceptualization:** Léa Duchesne, Gilles Hejblum, Karine Lacombe.

**Data curation:** Léa Duchesne.

**Formal analysis:** Léa Duchesne, Gilles Hejblum.

**Investigation:** Léa Duchesne, Gilles Hejblum, Richard Njouom, Coumba Touré Kane, Thomas d'Aquin Toni, Raoul Moh, Babacar Sylla, Nicolas Rouveau, Alain Attia, Karine Lacombe.

**Methodology:** Léa Duchesne, Gilles Hejblum.

**Software:** Léa Duchesne.

**Supervision:** Karine Lacombe.

**Validation:** Léa Duchesne, Gilles Hejblum.

**Visualization:** Léa Duchesne, Gilles Hejblum.

**Writing – original draft:** Léa Duchesne, Gilles Hejblum.

**Writing – review & editing:** Léa Duchesne, Gilles Hejblum, Richard Njouom, Coumba Touré Kane, Thomas d'Aquin Toni, Raoul Moh, Babacar Sylla, Nicolas Rouveau, Alain Attia, Karine Lacombe.

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
