## [Decision Letter · Decision Letter 0]

8 Jun 2020

PONE-D-20-07776

Model-based cost-effectiveness estimates of testing strategies for diagnosing hepatitis C virus infection in Central and Western Africa

PLOS ONE

Dear Dr. Duchesne,

Thank you for submitting your manuscript to PLOS ONE. After careful consideration, we feel that it has merit but does not fully meet PLOS ONE’s publication criteria as it currently stands. Therefore, we invite you to submit a revised version of the manuscript that addresses the points raised during the review process.

Your manuscript was reviewed by 2 experts in the field. Although both liked your paper, they identified several important problems, requiring your attention. Please consider the attached comments and provide point-by-point responses.

We look forward to receiving your revised manuscript.

Kind regards,

Yury E Khudyakov, PhD

Academic Editor

PLOS ONE

Journal Requirements:

"K. Lacombe reports personal fees and non-financial support from GILEAD, personal

fees and non-financial support from ABBVIE, personal fees and non-financial support

from JANSSEN, grants, personal fees and non-financial support from MSD, outside the

submitted work."

3. Please amend the manuscript submission data (via Edit Submission) to include authors Gilles Hejblum, Richard Njouom, Coumba Touré Kane, Thomas d’Aquin Toni, Raoul Moh, Babacar Sylla, Nicolas Rouveau, Alain Attia and Karine Lacombe.

Reviewers' comments:

Reviewer's Responses to Questions

**Comments to the Author**

1. Is the manuscript technically sound, and do the data support the conclusions?

Reviewer #1: Yes

Reviewer #2: Yes

2. Has the statistical analysis been performed appropriately and rigorously? 

Reviewer #1: Yes

Reviewer #2: Yes

3. Have the authors made all data underlying the findings in their manuscript fully available?

Reviewer #1: Yes

Reviewer #2: Yes

4. Is the manuscript presented in an intelligible fashion and written in standard English?

Reviewer #1: Yes

Reviewer #2: Yes

5. Review Comments to the Author

Reviewer #1: Thank you for conducting this model and reporting using the CHEERS guidance, it certainly helps in the review process. Please consider the following comments to help you through the peer-review process:

1. Health Sector Only - The 1st and 2nd Panels on Cost Effectiveness both recommend health economists take a societal perspective in CEA with the 2nd panel recommending to present both a "health sector" and a "societal" amount. Can the authors comment further on why items such as patient time, travel, and productivity were not considered?

2. Nomenclature - While the strategy names are logical and clear for a modeler, I'm worried the "average" reader will have a difficult time following the paper. Have you considered naming your strategies in a way that would be easier to read?

3. Smarter screening? - Physicians who determine an individual has a very low probability of disease (based on interview) can have a major impact on the effectiveness of screening. Applying a mass-population screening may be inappropriate for a variety of reasons. This variable would likely be the same for all 12 strategies, so it wouldn't impact the ICER, but can you comment on how your approach may bias the budget impact?

Reviewer #2: The authors present a thought provoking and rigorous study comparing the cost-effectiveness of 12 different testing strategies for HCV screening in LMICs with the base case data for this study specifically from Cameroon, Cote d'Ivoire, and Senegal. The manuscript is presented in an intelligible way and I commend the authors for their thorough approach with a robust sensitivity analysis. It is almost ready to accept as is; however, I have a few minor suggestions that I think would improve the manuscript.

1. The abstract is very hard to read prior to having read the paper and I believe that the results should be re-written without using the testing abbreviations as these can be quite confusing. Rather please use language to describe the different testing scenarios and maybe only highlight a few in the abstract.

2. I would like to see uncertainty estimates presented in figure 2 and in figure S1 and S2. I believe that these would help further improve the transparency of an already fantastic analysis.

3. I was not able to find the CHEERS statement among the additional files if this could be made available as a supplementary file to the manuscript that would be great.

4. If possible could all R code be made available for free either as supplementary files or through a link to a repository like GitHub or OSF.io

6. PLOS authors have the option to publish the peer review history of their article (what does this mean?). If published, this will include your full peer review and any attached files.

Reviewer #1: Yes: T. Joseph Mattingly II

Reviewer #2: No

---

## [Author Response · Author response to Decision Letter 0]

23 Jul 2020

Responses to the Journal requirements included in the notification e-mail for submission revision:

1. In this revised version of the manuscript, we attempted to meet all PLOS ONE's style requirements and hope that it will satisfy you.

2. As requested, an updated Competing Interests statement has been added to our cover letter.

3. All authors data have been added to the submission via the manuscript submission system.

---

## [Decision Letter · Decision Letter 1]

10 Aug 2020

Model-based cost-effectiveness estimates of testing strategies for diagnosing hepatitis C virus infection in Central and Western Africa

PONE-D-20-07776R1

Dear Dr. Duchesne,

We’re pleased to inform you that your manuscript has been judged scientifically suitable for publication and will be formally accepted for publication once it meets all outstanding technical requirements.

Kind regards,

Yury E Khudyakov, PhD

Academic Editor

PLOS ONE

Additional Editor Comments (optional):

Reviewers' comments:

Reviewer's Responses to Questions

**Comments to the Author**

1. If the authors have adequately addressed your comments raised in a previous round of review and you feel that this manuscript is now acceptable for publication, you may indicate that here to bypass the “Comments to the Author” section, enter your conflict of interest statement in the “Confidential to Editor” section, and submit your "Accept" recommendation.

Reviewer #2: All comments have been addressed

2. Is the manuscript technically sound, and do the data support the conclusions?

Reviewer #2: Yes

3. Has the statistical analysis been performed appropriately and rigorously? 

Reviewer #2: Yes

4. Have the authors made all data underlying the findings in their manuscript fully available?

Reviewer #2: Yes

5. Is the manuscript presented in an intelligible fashion and written in standard English?

Reviewer #2: Yes

6. Review Comments to the Author

Reviewer #2: Fantastic job I look forward to the piece being published. Congratulations to the author team on an excellent robust piece of science that moves the field forward.

7. PLOS authors have the option to publish the peer review history of their article (what does this mean?). If published, this will include your full peer review and any attached files.

Reviewer #2: No